# Application of Radiomics for Differentiating Lung Neuroendocrine Neoplasms

**DOI:** 10.3390/diagnostics15070874

**Published:** 2025-03-31

**Authors:** Aleksandr Borisov, David Karelidze, Mikhail Ivannikov, Elina Shakhvalieva, Peri Sultanova, Kirill Arzamasov, Nikolai Nudnov, Yuriy Vasilev

**Affiliations:** 1Research and Practical Clinical Center for Diagnostics and Telemedicine Technologies of the Moscow Health Care Department, 127051 Moscow, Russia; arzamasovkm@zdrav.mos.ru (K.A.); vasilevya1@zdrav.mos.ru (Y.V.); 2Department of Medical Cybernetics and Informatics, Pirogov Russian National Research Medical University, 117513 Moscow, Russia; 3Russian Scientific Center of Roentgenoradiology, 117997 Moscow, Russia; david_ka@mail.ru (D.K.); sulperi14@mail.ru (P.S.); nudnov@rncrr.ru (N.N.); 4City Clinical Hospital named after A.K. Yeramishantsev, 129327 Moscow, Russia; ivannikovmichail@gmail.com; 5Children’s City Clinical Hospital No. 9 named after G.N.Speransky, 123317 Moscow, Russia; shelina9558@gmail.com; 6Department of Artificial Intelligence Technologies, MIREA—Russian Technological University, 119454 Moscow, Russia

**Keywords:** radiomics, neuroendocrine neoplasms, lung cancer, machine learning

## Abstract

**Background/Objectives:** Lung neuroendocrine neoplasms (NENs) are a heterogeneous group of tumors requiring accurate differentiation from non-small cell lung cancer (NSCLC) for effective treatment. Conventional computed tomography (CT) lacks pathognomonic features to distinguish these subtypes. Radiomics, which extracts quantitative imaging features, offers a potential solution. **Methods:** This retrospective multicenter study included 301 patients with histologically confirmed lung cancer who underwent native CT scans. The dataset comprised 150 NSCLC cases (75 adenocarcinomas, 75 squamous cell carcinomas) and 151 NENs (75 SCLC, 60 carcinoids, 16 large cell neuroendocrine carcinomas). Tumors were manually segmented, and 107 radiomics features were extracted. Dimensionality reduction and feature selection were performed using Pearson correlation analysis and LASSO regression. Decision tree and random forest classifiers were trained and evaluated using a 70:30 training–testing split. Model performance was assessed using the area under the receiver operating characteristic curve (AUC), accuracy, precision, recall, and F1-score. **Results:** The model differentiating NENs from NSCLC achieved an AUC of 0.988 on the test set, with an accuracy of 97.8%. The model distinguishing SCLC from other NENs attained an AUC of 0.860 and an accuracy of 82.6%. First-order and textural radiomics features were key discriminators. **Conclusions:** Radiomics-based machine learning models demonstrated high diagnostic accuracy in differentiating lung NENs from NSCLC and in subclassifying NENs. These findings highlight the potential of radiomics as a non-invasive, quantitative tool for lung cancer diagnosis, warranting further validation in larger multicenter studies.

## 1. Introduction

Lung cancer (LC) is a term that encompasses a variety of malignant epithelial tumors [1]. Major histologic types include a group of non-small cell carcinomas and neuroendocrine neoplasms [2]. Non-small cell lung cancer (NSCLC) represents a group of malignant neoplasms that includes several histologic types, such as adenocarcinoma (LADC), squamous cell carcinoma (SCC), and large cell carcinoma (LCC) [3]. In the majority of patients (60–80%) with newly diagnosed NSCLC, the disease is diagnosed at stage III–IV, which requires chemotherapy, but at stage I–IIIA, treatment can be radical due to the possibility of surgical treatment [4]. A completely different approach to treatment is warranted for neuroendocrine neoplasms.

Neuroendocrine neoplasms (NENs) are a heterogeneous group of tumors that arise from neuroendocrine cells [5]. Typically, NENs are classified into two distinct categories: less aggressive neuroendocrine tumors (NETs) and neuroendocrine carcinomas (NECs). These two categories exhibit significant morphologic and clinical differences [6]. According to the 2021 World Health Organization (WHO) classification, NETs encompass typical carcinoid (TC), atypical carcinoid (AC), and carcinoid tumors with a high proliferation index. NECs, on the other hand, encompass small cell lung carcinoma (SCLC) and large cell neuroendocrine carcinoma (LCNEC). Notably, NENs comprise 20% of all lung neoplasms, with 15% of these cases classified as SCLC [6,7]. The primary distinguishing feature among these histologic types is the mitotic index, defined as the number of mitoses per 2 мм2, and/or the presence of areas of necrosis within the tumor mass. The greater the observed mitoses, the higher the grade and the poorer the prognosis [8]. Notably, carcinoid tumors are more frequently detected at an early stage, which allows for surgical intervention in the form of a lobectomy. The five-year survival rate for low-grade carcinoid is 88–89% [9,10]. In contrast, SCLC necessitates a distinctly different array of treatment strategies due to its high degree of malignancy and poor prognosis. In most cases, SCLC is detected at an advanced stage, chemotherapy plays a leading role in treatment, and the five-year survival rate can reach up to 9.1% [11,12].

Therefore, an early and accurate differential diagnosis is essential for effective management strategies. Computed tomography (CT) is a primary imaging modality in the radiologic diagnosis of lung cancer; however, there are no pathognomonic CT signs that can reliably distinguish between different histologic types of neoplasms (Figure 1) [13,14,15]. The histotype of a neoplasm can only be definitively ascertained through biopsy or postoperative histologic verification.

Radiomics, a method of extracting and analyzing quantitative parameters from medical diagnostic images, has demonstrated significant potential in differentiating between various histologic types of lung neoplasms [16]. However, there is a paucity of studies dedicated to the differential radial diagnosis of NEN, NSCLC, and SCLC. Early research by Cozzi et al. identified Skewness and ClusterShade features in CT scans as potential biomarkers for NEN grade and metastases, though findings were constrained by small sample sizes [17]. Bicci et al. later expanded the cohort and identified different significant features, underscoring the need for larger studies [18].

Further research by Thuillier et al. [19] found that adding radiomic features to PET/CT models did not significantly improve classification, while Adelsmayr et al. [20] and Liu et al. [21] demonstrated the potential of radiomics in distinguishing NENs from other lung neoplasms using machine learning models. However, small sample sizes and the exclusion of certain lung cancer histotypes limit generalizability.

In this study, we sought to address previously understudied issues in this field. Our work represents one of the largest cohorts examined to date and benefits from standardized scanning protocols. Another unique element of our work is the use of two classification models: a model that separates neuroendocrine neoplasms from the most common types of non-small cell lung cancer and a model that separates small cell lung cancer from other neuroendocrine neoplasms.

## 2. Materials and Methods

The purpose of this work is to investigate the possibilities of radiomics for the differential diagnosis of neuroendocrine lung neoplasms based on native CT scans. At the first stage of the study, we searched for CT scans of patients with lung cancer, primarily with neuroendocrine neoplasms. Then we segmented the region of interest on CT and extracted radiomics features from it. The end point of the study was the construction of machine learning models that differentiate various NEN groups from NSCLC and from each other.

### 2.1. Patients’ Selection

Patients who were followed up with a primary diagnosis of lung cancer in the medical facilities of the Moscow Healthcare Department (MHCD) and Budgetary Institution Russian Scientific Center of Roentgenoradiology (RSCRR) of the Ministry of Healthcare of the Russian Federation between 2014 and 2022 were considered for this retrospective study. The selection of patients was based on the analysis of electronic medical records of the patient containing the basic diagnosis code C34 <<Malignant neoplasm of bronchia and lung>> according to the International Classification of Diseases (ICD), excluding code 34.0 <<Malignant neoplasm: Main bronchus>>. The inclusion criteria were the age of patients being over 18 years old, the presence of histologically confirmed lung cancer using histological study and/ or immunohistochemistry (the results of cytological studies were not used to determine the type of lung cancer due to the low degree of consistency of these studies [22], especially with small sizes of tumors), and the presence of a thin-slice CT scan before treatment. To increase the reproducibility of the study results and offset differences related to the scanning protocol, the study used a native series of thin-slice CT scans of the chest with a slice thickness of 1.00 mm and a tube voltage of 120 kV (which was the most common scanning protocol in the population we evaluated). The list of CT scanner models included in the study is as follows: Toshiba Aquilion 64, Toshiba Aquilion CXL 128, Toshiba Aquilion ONE 320, Toshiba Alexion TSX-032A, Toshiba Aquilion Prime, Siemens Somatom Definition, Siemens Somatom Perspective, Siemens Somatom Definition AS, Philips Ingenuity CT, Philips Brilliance CT 64, Philips Ingenuity Elite 128, General Electric Revolution EVO, and the GE Medical Systems Brightspeed. The search for CT scans was performed in the PACS system of the RSCRR and in the Unified Radiological Information Service of Moscow City [23], using the Dataset curation platform [24]. The exclusion criteria were as follows: treatment before histological verification of the masses, ambiguous results of immunohistochemistry, benign lung neoplasms according to histology, poor-quality CT images, and metachronous oncological processes.

The detailed process of patient selection is presented in Figure 2.

The initial sample included 3000 patients; of these, 1678 had no data on histological or immunohistochemical verification of the diagnosis and 1046 patients were unable to receive a CT scan before treatment. As a result, the sample was reduced to 632 people. The structure of this sample consisted of 249 lung adenocarcinomas (LADC), 180 squamous cell lung cancers (SCC), 127 small cell lung cancer (SCLC), 34 typical carcinoids, 26 atypical carcinoids, and 16 large cell neuroendocrine carcinomas (LCNEC). First of all, we were looking for patients with neuroendocrine neoplasms, so the proportions of patients with various types of lung cancer that we selected may differ from those in the population. In this study, due to their small number, large cell neuroendocrine carcinomas and carcinoid tumors were grouped into one group as non-small cell lung cancer with neuroendocrine differentiation. In order to avoid class imbalance when building machine learning models, for large groups we selected random studies based on the number of objects in the smallest group (LCNEC + carcinoids). The final sample consists of 76 LCNEC + carcinoids, 75 small cell lung cancers, 75 lung adenocarcinomas, and 75 squamous cell lung cancers.

### 2.2. Radiomics Feature Extraction and Radiological Signs Assessment

All studies were reviewed by three radiologists, with 5 years of experience in thoracic imaging. The entire volume of the tumor was manually segmented on native CT scans using the 3DSlicer software, version 5.4.0 [25]. No image preprocessing was carried out. Standardization of the images was carried out by choosing a single scanning protocol and slice thickness. In the event of contrast-enhanced scans, radiologists could use them to refine tumor segmentation. Masks obtained by one radiologist were reviewed by another radiologist to eliminate segmentation errors. In cases of disagreement, the decision was made through a collective discussion. The segmentation criteria for the radiologist in our study were as follows: segmentation was performed in a soft-tissue window; we segmented the entire volume of the tumor along its border; we ensured the exclusion of atelectasis from the ROI; if possible, we avoided the inclusion in the ROI of bronchi and blood vessels; we excluded large cavities of destruction and bronchial lumens from the ROI; we did not exclude calcifications; we evaluated the boundaries of the tumor on post-contrast series, if available; in the case of several tumor foci, one of the largest or most differentiated foci from the surrounding tissues was segmented; if the tumor was large and it was impossible to reliably determine its boundaries with surrounding tissues, we were allowed to segment a limited area of the tumor that reliably related specifically to it.

Textural feature extraction was carried out through the SlicerRadiomics plugin based on the PyRadiomics version 3.1.0 library. A total of 107 features of the PyRadiomics lists were selected, belonging to the first-order, shape-based 3D, gray level co-occurrence matrix (GLCM), gray level size zone matrix (GLSZM), gray level run length matrix (GLRLM), neighboring gray tone difference matrix (NGTDM), and gray level dependence matrix (GLDM) classes.

During the segmentation of the tumor, radiologists also assessed the localization (the lung and its lobe) and radiological signs of the neoplasm, such as the spiculation, pleural indentation, and presence of decay cavities.

### 2.3. Model Development

The Python programming language version 3.10 and the standard libraries in this language used for data analysis and machine learning (scikit-learn, pandas, numpy, lightgbm, matplotlib) were used to build machine learning models. In this work, we have developed two classification models: a model that separates neuroendocrine neoplasms from the most common types of non-small cell lung cancer (adenocarcinomas + squamous cell cancer) and a model that separates small cell lung cancer from other neuroendocrine neoplasms. To reduce the dimension, correlation analysis using the Pearson correlation coefficient was used, and the significance of each feature was assessed using the feature_importances measure based on the forest of trees and LASSO regression. After selecting the most significant features, the dataset was divided into training and test samples in a 70:30 ratio. Decision tree, random forest, logistic regression, and gradient boosting classifiers were used to build machine learning models, and optimal hyperparameters for these classifiers were selected using the grid-search method. The following parameters were selected for the decision tree: criterion, maximum depth, and the minimum number of samples in a leaf node. For the random forest, the following parameters were selected: number of trees in the forest, maximum depth of the tree, and the minimum number of samples in a leaf node. For the gradient boosting, the following parameters were selected: number of trees in the forest, maximum depth of the tree, the minimum number of samples in a leaf node, and learning_rate. For the logistic regression, the following parameters were selected: penalty, solver, and the maximum number of iterations.

### 2.4. Statistical Analyses

Continuous variables were presented as medians and interquartile ranges (IQR), analyzed using the Mann–Whitney U test for group comparisons. Categorical variables were presented as frequencies and percentages, and their group comparisons were conducted by Pearson’s chi-squared test. The significance threshold was set at p = 0.05. The model’s performance was evaluated using the area under the receiver operating characteristic curve (AUC), precision, recall, accuracy, and F1-score metrics. The Clopper–Pearson exact method was used for calculating the 95% binomial confidence intervals. Statistical analysis was performed with Python (version 3.10) and SPSS software (version 23.0).

When determining the sample size, we were guided by the maximum number of non-small cell neuroendocrine neoplasms that we could find. The number of studies in the larger groups corresponded to the number of studies in this smallest group to avoid class imbalance when training models.

## 3. Results

### 3.1. Patient’s Characteristics

#### 3.1.1. Model for Differentiating NENs from NSCLC

A total of 301 patients (median age, 66 years [IQR, 59–71 years]) were included in the first model. Of these, there were 194 (64.5%) men and 107 (35.5%) women. These were 150 patients with NSCLC (75 with lung adenocarcinoma and 75 with squamous cell cancer) and 151 patients with NENs (60 with carcinoids, 16 with large cell neuroendocrine carcinomas, and 75 with small cell lung cancer). All the baseline characteristics are detailed in Table 1.

On average, patients in the NENs group were slightly younger, and there were more women present than in the NSCLC group. In both groups, neoplasms were more common in the right lung. However, the location in the lower and middle lobes was more typical for the NENs group than for the NSCLC group, in which the tumors were mainly located in the upper lobe. Compared with the NSCLC group, the NENs group exhibited significantly lower occurrences in the spiculation, pleural indentation, and presence of decay cavities.

#### 3.1.2. Model for Differentiating SCLC from Other NENs

A total of 151 patients (median age, 64 years [IQR, 57–70 years]) were included in the second model. Of these, there were 89 (58,9%) men and 62 (41,1%) women. There were 75 patients with SCLC and 76 patients with NENs (34 typical carcinoids, 26 atypical carcinoids, and 16 with large cell neuroendocrine carcinomas). All the baseline characteristics are detailed in Table 2.

There were no age differences between the groups. However, there were more women than men in the other NENs group, while in SCLC group there were more men. SCLC tumors were mainly located in the left lung and preferred the upper lobe, unlike in the other NENs group. Also, a large proportions of spiculation and pleural indentation are typical for SCLC group.

### 3.2. Model Construction

#### 3.2.1. Model for Differentiating NENs from NSCLC

Statistically significant differences were observed in 86 of the considered features. After reducing the dimension, five radiomics features were selected (two first-order features and three s-order features), namely firstorder_Minimum, ngtdm_Strength, glszm_SizeZoneNonUniformityNormalized, firstorder_Energy, and glcm_Correlation. The performance of classification models is shown in Table 3. The best classification results were shown by the decision tree model with a depth of 3. The receiver operating characteristic curve analysis of the best model for differentiating lung neuroendocrine neoplasms from non-small cell lung tumors in the training set and test set is shown in Figure 3. The plotting decision tree model is shown in Figure 4.

The firstorder_Minimum feature shows that in the NENs group, the minimum density value is, on average, 4 times lower than that of NSCLC, which may indicate a larger number of areas of airiness and small cavities inside the tumor. The ngtdm_Strength feature is on average an order of magnitude higher in the NENs group, which suggests that these masses have slow change in intensity but more large coarse differences in gray level intensities. This may indicate the presence of large heterogeneous sites within the NENs. The Glszm_SizeZoneNonUniformityNormalized feature indicates more homogeneity among zone size volumes in NSCLC tumors. The glcm_Correlation feature shows a higher linear dependency of gray level values to their respective voxels in NENs.

The model we have developed shows high diagnostic accuracy metrics, which makes it possible to determine NENs using CT data with high accuracy. Adding radiological and localization signs to the feature space did not increase the accuracy metrics of this model.

#### 3.2.2. Model for Differentiating SCLC from Other NENs

Statistically significant differences were observed in 82 of the considered features. After reducing the dimension, seven radiomics features were selected (one first-order feature, two shape-based features, and four s-order features), namely ngtdm_Strength, glszm_SizeZoneNonUniformityNormalized, shape_MajorAxisLength, firstorder_Kurtosis, shape_LeastAxisLength, glszm_GrayLevelVariance, and glszm_ZoneEntropy. The performance of the classification models is shown in Table 4. The best classification results were shown by the random forest model, with three trees and a tree depth of five. The receiver operating characteristic curve analysis of best model for differentiating SCLC from other NENs in the training set and test set is shown in Figure 5. Adding radiological and localization signs to the feature space also did not increase the accuracy metrics of this model.

The ngtdm_Strength index in the SCLC group is, on average, 3 times lower than in the other NENs group, which indicates that the SCLC texture contains fewer large differences areas of gray levels. Shape-based features, such as shape_MajorAxisLength and shape_LeastAxisLength, indicate that SCLC tumors are characterized by larger sizes on average. The glssm_GrayLevelVariance feature indicates that SCLC is characterized by less variance in grey level in the image. SCLC is also characterized by higher values of first order_Kurtosis, which suggests that the mass of the distribution of SCLC is concentrated towards the tail(s) rather than towards the mean.

## 4. Discussion

A survey of the scientific literature reveals a paucity of studies in the field of NEN radiomics. One of the earliest works in this area is the study by D. Cozzi et al., which focused on differentiating various histologic subtypes of NENs based on Ki-67 expression levels [17]. Ki-67, a nuclear antigen produced by proliferating cells, has been shown to correlate with the grade of NEN. In their study, the authors performed radiomic analysis of native and contrast CT-series in 27 patients with NEN of different grades. The analysis of contrast series revealed statistically significant differences in the Skewness and ClusterShade features. These features are used to characterize the degree of heterogeneity of the mass. Furthermore, the authors observed that these features exhibited statistically significant differences between patients with and without NEN metastases. The authors conclude that the Skewness and ClusterShade features have the potential to serve as biomarkers, indicating the grade and aggressiveness of NEN. It is important to acknowledge the limitations of this study, as well as other studies in the field of NEN radiomics, which are imposed by the relatively small sample size, an effect of the low incidence of neuroendocrine neoplasms in the population. It is also important to note that the grading neoplasms by Ki-67 level is not currently practiced in the WHO classification of NENs [8].

One year after the publication of the previous study, the authors, under the direction of E. Bicci, released a subsequent paper employing the same methods, but with an expanded sample of 60 patients [18]. This subsequent study yielded statistically significant differences in native CT-series, particularly in the Maximal correlation coefficient, ClusterProminence, and Strength features. Notably, the features obtained in the authors’ previous study, namely Skewness and ClusterShade, were not among the seven statistically significantly different features in the contrast CT-series. This discrepancy highlights the necessity for additional research with increasing sample sizes and enhanced data preparation techniques, such as resampling, to minimize variability among the features.

Another study that focused on the differentiation of histotypes of NEN is the work of P. Thuillier et al. [19]. In this study, the authors divided a sample of 46 patients with NEN into groups with NETs and NECs. Subsequently, they evaluated the efficacy of classification models based on PET/CT features (SUV max, SUV mean, etc.) and radiomics features. The authors concluded that, in the case of PET/CT, the incorporation of radiomics features into the classification model did not result in a significant change. It should be noted that the utilization of PET/CT features fell outside the scope of the present study. This area requires further study to clarify its implications.

The work of G. Adelsmayr et al. exhibited a high degree of similarity to that of our own [20]. The authors divided a sample of 133 patients with histologically verified neoplasms into subgroups with LADC, SCLC, SCC, and NET. A distinctive aspect of their study was the extraction of radiomics features from standard native CT series and CT series with a −50 HU threshold. The GLSZM_SmallAreaEmphasis score exhibited the highest ROC-AUC of 0.818 when differentiating NETs from all other neoplasms. Furthermore, GLCM_Correlation exhibited a ROC-AUC of 0.81, demonstrating its statistical significance in differentiating NETs from SCLC. The findings of this study are consistent with those of our own, as they demonstrate that radiomics features exhibit significant differences between the groups of patients with NSCLC, SCLC, and NETs.

A recent study by X. Liu et al. also demonstrates notable parallels with our own research [21]. This multicenter study encompassed 445 patients with peripheral lung solid nodules who had been verified to have NEN and adenocarcinoma. Within the NEN cohort, 38 patients were diagnosed with NETs, 41 with LCNECs, and 122 with SCLC. Notably, this study stands out for its focus on rare tumors and for its attempt to analyze visual CT features to distinguish different histologic cancer types. It then combined these features with radiomic features to create a unified model. The study’s methodology involved the development of the following three machine learning models: (1) a model based on CT radiological features with an ROC-AUC of 0.729; (2) a model based on radiomic features with an ROC-AUC of 0.787; and (3) a combined model with an ROC-AUC of 0.807. While the machine learning models exhibited high classification accuracy for the designated tumor subtypes, the study’s sample size posed a substantial limitation. The study’s lack of patients with other significant histological types of lung cancer restricted its external validity. Nevertheless, the study emphasized the potential applicability of radiomic analysis in the differential diagnosis of lung cancer histologic types.

The scientific literature includes works devoted to the differentiation of NEN and hamartomas [26,27,28]. This is due to the fact that both neoplasms have similar morphology and CT signs but require radically different approaches in therapy. The differentiation of NEN and hamartomas was not included in the objectives of our study.

It should be noted that the present study is subject to several limitations. The most significant limitation is the modest sample size, which is attributable to the relatively infrequent occurrence of NENs within the studied population. With a further increase in the sample, the set of the most significant features, the most effective model, or its accuracy metrics may change. Nevertheless, we were able to collect one of the largest samples of NENs, second in size only to the sample of X. Liu. The results of the study by D. Cozzi et al. suggest the possibility of using contrast-enhanced CT series; however, we deliberately declined to use them because of the infrequent use of contrast enhancement in routine chest CT scans. It is noteworthy that our research, to the best of our knowledge, is among the first multicenter studies, and the models we created have the highest ROC-AUC values. It is noteworthy that the high values obtained in this study may be attributable to the standardization of the scanning protocols examined in this paper. We deliberately excluded patients who were analyzed using alternative scanning protocols, thereby enhancing the reproducibility of our models. We plan to continue this research in the future. We want to continue the search for patients with this pathology, and to use additional radiological features, including those based on wavelet transforms and Log kernel sizes. We believe that the development of radiomics is dependent on the standardization of scanning protocols and the post-processing of data [29,30]. The field of NEN radiomics remains largely unexplored territory that necessitates further investigation through larger, multicenter studies.

## 5. Conclusions

Neuroendocrine lung neoplasms are a rare pathology that require a special approach to diagnostics and treatment. Radiomics is a promising area of research in this field. Our CT radiomics models demonstrated effective performance in distinguishing between NETs and NSCLC with an AUC of 0.988 [0.940; 0.999] and between SCLC and other NETs with an AUC of 0.860 [0.732; 0.950]. Therefore, when standardizing extraction and processing processes, the radiomics model may serve as a non-invasive, quantitative, objective, and sensitive approach for differentiating different types of lung cancer.

## Figures and Tables

**Figure 1 diagnostics-15-00874-f001:**
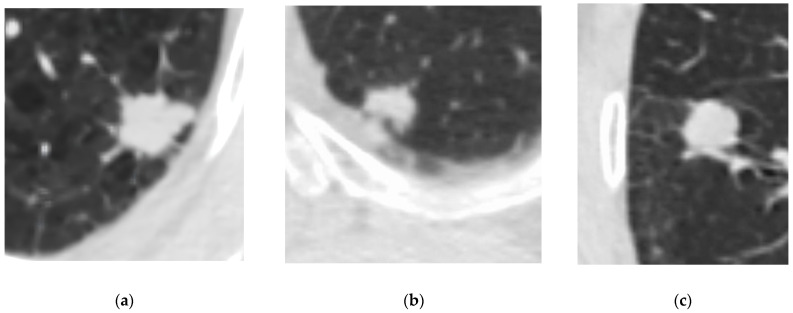
Peripheral lung masses of different lung cancer subtypes that cannot be distinguished by CT features: (**a**) peripheral non-small cell adenocarcinoma, (**b**) small cell lung cancer, and (**c**) carcinoid tumor of the lung.

**Figure 2 diagnostics-15-00874-f002:**
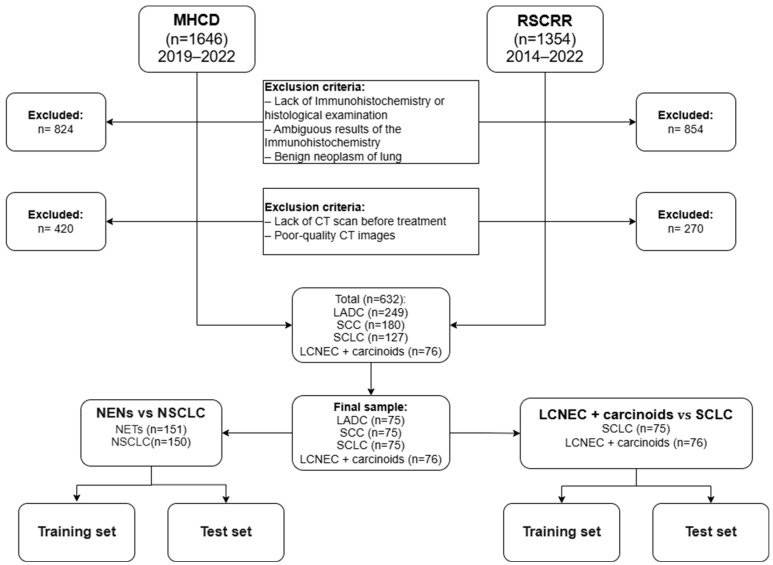
Flow diagram of the patient selection. MHCD indicates the Moscow Healthcare Department; RSCRR indicates the Russian Scientific Center of Roentgenoradiology; LADC, lung adenocarcinoma; SCC, squamous cell cancer; SCLC, small cell lung cancer; LCNEC, large cell neuroendocrine carcinoma; NENs, neuroendocrine neoplasms; NSCLC, non-small cell lung cancer.

**Figure 3 diagnostics-15-00874-f003:**
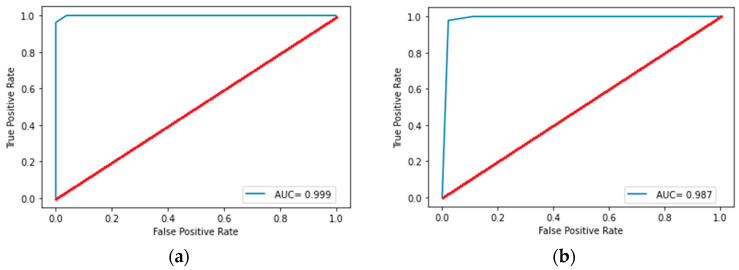
Receiver operating characteristic curve analysis of the best model for differentiating lung neuroendocrine neoplasms from non-small cell lung tumors in the training (**a**) set and test (**b**) set. AUC, area under the receiver operating characteristic curve.

**Figure 4 diagnostics-15-00874-f004:**
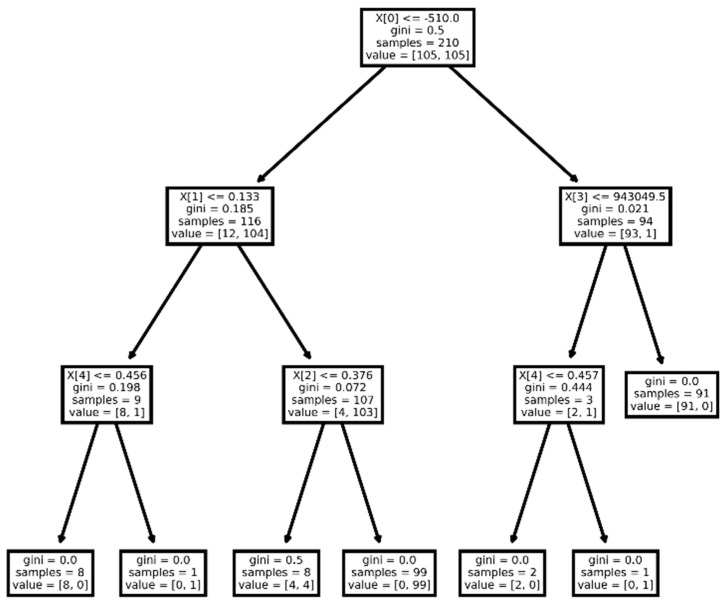
Visualization of decision tree for differentiating lung neuroendocrine neoplasms from non-small cell lung tumors. X[0]—firstorder_Minimum feature, X[1]—ngtdm_Strength feature, X[2]—glszm_SizeZoneNonUniformityNormalized feature, X[3]—firstorder_Energy feature, and X[4]—glcm_Correlation feature.

**Figure 5 diagnostics-15-00874-f005:**
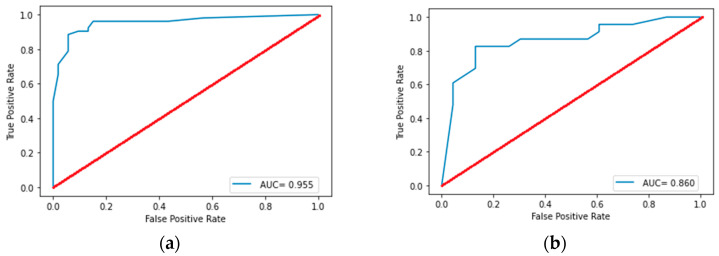
Receiver operating characteristic curve analysis of the best model for differentiating SCLC from other NENs in the training (**a**) set and test (**b**) set. AUC, area under the receiver operating characteristic curve.

**Table 1 diagnostics-15-00874-t001:** Baseline patient characteristics in the set for model for differentiating NENs from NSCLC.

Characteristic	Total (*n* = 301)	NSCLC (*n* = 150)	NENs (*n* = 151)	*p*-Value
Age (y)	66.0 [59.0; 71.0]	67.0 [62.7; 72.0]	64.0 [57.0; 70.0]	**0.002**
Sex (male)	194 (65.5%)	105 (70.0%)	89 (58.9%)	0.054
Lung (right)	185 (61.5%)	93 (62.0%)	92 (60.9%)	0.565
Lobe (upper)	155 (51.5%)	92 (61.3%)	63 (41.7%)	**0.001**
Lobe (lower)	118 (39.2%)	50 (33.3%)	68 (45.0%)	**<0.001**
Lobe (middle)	28 (9.3%)	8 (5.3%)	20 (13.3%)	**0.004**
Spiculation	204 (67.8%)	133 (88.7%)	71 (47.0%)	**<0.001**
Pleural indentation	201 (66.8%)	129 (86.0%)	72 (47.7%)	**<0.001**
Decay cavities	25 (8.3%)	22 (14.7%)	5 (3.3%)	**0.001**

**Table 2 diagnostics-15-00874-t002:** Baseline patient characteristics in the set for model for differentiating SCLC from other NENs.

Characteristic	Total (*n* = 151)	Other NENs (*n* = 76)	SCLC (*n* = 75)	*p*-Value
Age (y)	64 [57.0; 70.0]	64.0 [57.0; 70.0]	64.0 [57.0; 71.0]	0.681
Sex (male)	89 (58.9%)	29 (38.2%)	60 (80.0%)	**<0.001**
Lung (right)	92 (60.1%)	57 (75.0%)	35 (46.7%)	**0.001**
Lobe (upper)	63 (41.7%)	23 (30.1%)	40 (53.3%)	**0.001**
Lobe (lower)	68 (45.0%)	37 (48.7%)	31 (41.3%)	0.365
Lobe (middle)	20 (13.2%)	16 (21.1%)	4 (5.3%)	**0.004**
Spiculation	71 (47.0%)	21 (27.6%)	50 (66.7%)	**<0.001**
Pleural indentation	72 (47.7%)	30 (39.5%)	42 (56.0%)	**<0.001**
Decay cavities	5 (3.3%)	2 (2.6%)	3 (4.0%)	0.681

**Table 3 diagnostics-15-00874-t003:** Diagnostic performance of models for differentiating NENs from NSCLC.

		AUC	Precision	Recall	F1-Score	Accuracy
Decision tree	Training set	0.999 [0.983; 1.00]	1.00 [0.983; 1.00]	0.962 [0.926; 0.983]	0.981 [0.952; 0.995]	0.981 [0.952; 0.995]
	Test set	0.988 [0.940; 0.999]	0.978 [0.922; 0.997]	0.978 [0.922; 0.997]	0.978 [0.922; 0.997]	0.978 [0.922; 0.997]
Random forest	Training set	0.997 [0.974; 1.00]	0.958 [0.920; 0.980]	0.986 [0.959; 0.997]	0.972 [0.939; 0.989]	0.967 [0.933; 0.987]
	Test set	0.966 [0.906; 0.993]	0.967 [0.906; 0.993]	0.967 [0.906; 0.993]	0.967 [0.906; 0.993]	0.962 [0.905; 0.993]
Logistic regression	Training set	0.997 [0.974; 1.00]	0.979 [0.952; 0.995]	0.986 [0.959; 0.997]	0.982 [0.952; 0.995]	0.980 [0.952; 0.995]
	Test set	0.961 [0.891; 0.989]	0.950 [0.891; 0.989]	0.934 [0.861; 0.975]	0.942 [0.875; 0.982]	0.934 [0.861; 0.975]
Gradient boosting	Training set	0.999 [0.983; 1.00]	0.972 [0.939; 0.989]	1.00 [0.983; 1.00]	0.986 [0.959; 0.997]	0.984 [0.959; 0.997]
	Test set	0.972 [0.906; 0.993]	0.952 [0.891; 0.989]	0.967 [0.905; 0.993]	0.959 [0.891; 0.989]	0.953 [0.891; 0.989]

**Table 4 diagnostics-15-00874-t004:** Diagnostic performance of models for differentiating SCLC from other NENs.

		AUC	Precision	Recall	F1-Score	Accuracy
Random forest	Training set	0.955 [0.892; 0.984]	0.862 [0.786; 0.925]	0.962 [0.905; 0.990]	0.909 [0.832; 0.953]	0.904 [0.832; 0.953]
	Test set	0.860 [0.732; 0.950]	0.826 [0.680; 0.920]	0.826 [0.680; 0.920]	0.826 [0.680; 0.920]	0.826 [0.680; 0.920]
Decision tree	Training set	0.726 [0.631; 0.809]	0.653 [0.552; 0.741]	0.942 [0.881; 0.979]	0.771 [0.682; 0.849]	0.724 [0.631; 0.809]
	Test set	0.739 [0.581; 0.854]	0.657 [0.511; 0.800]	1.00 [0.921; 1.00]	0.793 [0.654; 0.904]	0.739 [0.581; 0.854]
Logistic regression	Training set	0.853 [0.767; 0.911]	0.755 [0.662; 0.833]	0.769 [0.682; 0.849]	0.762 [0.672; 0.841]	0.762 [0.672; 0.841]
	Test set	0.747 [0.605; 0.871]	0.667 [0.511; 0.800]	0.609 [0.443; 0.743]	0.636 [0.488; 0.781]	0.652 [0.488; 0.781]
Gradient boosting	Training set	0.989 [0.949; 1.00]	0.944 [0.811; 0.979]	0.981 [0.934; 0.998]	0.962 [0.906; 0.990]	0.962 [0.906; 0.990]
	Test set	0.685 [0.534; 0.818]	0.680 [0.534; 0.818]	0.739 [0.581; 0.854]	0.708 [0.557; 0.836]	0.696 [0.534; 0.818]

## Data Availability

The data representing the radiomics features are available upon request from the corresponding author. Initial CT scans are not available due to the protection of patients’ personal data.

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
