# Peer review of "Application of Radiomics for Differentiating Lung Neuroendocrine Neoplasms"

_diagnostics, 2025, doi:10.3390/diagnostics15070874_

Round 1

Reviewer 1 Report

Comments and Suggestions for Authors

Dear Authors,

I hope this message finds you well. Below are my comments and suggestions regarding the manuscript:

  1. The introduction does not include a review of previous studies. It is crucial to provide a detailed and comprehensive overview of relevant prior work.

  2. The introduction lacks any mention of studies on the use of radiomics. This should be addressed.

  3. One of the critical steps in studies utilizing medical images is image preprocessing. Unfortunately, your study does not refer to any preprocessing steps used. These should be added with precise details.

  4. Have Fourier transform-based radiomic features been used in this study? Previous studies have shown that Fourier-based radiomic features are highly valuable and significant.

  5. The study indicates that experts were involved in drawing ROI. However, no criteria for delineating these regions are specified. Clear and accurate criteria should be included in the study.

  6. In Figure 3, the AUC is approximately 1, indicating that your model performs exceptionally well and produces results similar to your gold standard. In your opinion, could this indicate overfitting? What measures have you taken to prevent overfitting?

  7. What is the gold standard for classifying the images and patients in your study? Is this gold standard based on radiologist observations? If a radiologist can classify tumor types simply by looking at the images, what is the necessity for conducting a radiomics study?

  8. In Figure 5, the ROC curve appears to be generated using a very small number of patients. Could you provide an explanation for this?

I hope these comments are helpful in improving the manuscript.

Best regards.

Reviewer 2 Report

Comments and Suggestions for Authors

The authors used radiomics features and two machine learning models to distinguish lung neuroendocrine neoplasms (NENs) from non-small cell lung cancer (NSCLC) and small cell lung cancer (SCLC) from NENs.

The introduction section should be extended to clearly present the paper's aim and the differences from recent studies.

The authors should provide a 1-2 paragraph explanation for the machine learning models considered in this study.

In Figure 2, there are "CV training set" and "CV test set". The "CV" is not explained in text or abbreviations. Since the authors used the hold-out method (70/30), it should not be "cross-validation".

Tables 3 and 4 present the results of Decision Tree and Random Forest, respectively, while these models achieved higher scores. However, the authors could add both models' results to inform the readers about the differences in the obtained results.

Additionally, even though obtaining superior results by two different models in two different experiments is common, it is a little bit confusing. This could decrease the possibility of clinical implementation. The authors need to discuss these points, or they might be added as a limitation.

In the experimental part, the authors choose a different number of features to train the models. The selection procedure is well-explained; however, what will be the results if the authors add or remove some features? Have any experiments performed on these features to demonstrate that they produced superior results?

There are several parameters in machine learning models, particularly in Random Forest. How did the authors decide these parameters (# of trees = 3, depth = 5, etc.)? Did the authors use grid-search?

Existing other tree-based machine learning algorithms, such as Gradient Boosting and Extreme Gradient Boosting, might produce higher scores since they have more advanced optimization. The author might focus on these models in their future work.

Round 2

Reviewer 2 Report

Comments and Suggestions for Authors

Thanks to the authors for addressing my concerns in the revised version.